# Correspondence Rules for *SU*(1,1) Quasidistribution Functions and Quantum Dynamics in the Hyperbolic Phase Space

**DOI:** 10.3390/e24111580

**Published:** 2022-10-31

**Authors:** Miguel Baltazar, Iván F. Valtierra, Andrei B. Klimov

**Affiliations:** Departamento de Física, Universidad de Guadalajara, Guadalajara 44420, Mexico

**Keywords:** phase space, Wigner function, *SU*(1,1) group

## Abstract

We derive the explicit differential form for the action of the generators of the SU(1,1) group on the corresponding *s*-parametrized symbols. This allows us to obtain evolution equations for the phase-space functions on the upper sheet of the two-sheet hyperboloid and analyze their semiclassical limits. Dynamics of quantum systems with SU(1,1) symmetry governed by compact and non-compact Hamiltonians are discussed in both quantum and semiclassical regimes.

## 1. Introduction

Representing non-linear quantum dynamics as an evolution of phase-space distributions not only offers an appealing visualization of sophisticated processes but also provides a convenient way to study the quantum–classical transition from the dynamical point of view [1,2,3]. The starting point for such analysis is the Liouville-like equation of motion for a quasidistribution $W_{\rho }\left(\zeta \right)$, which is a one-to-one map [4,5,6,7,8,9,10], of the density matrix $\hat{\rho }$ into a function defined on the classical phase space M,
$$\hat{\rho }\Leftrightarrow W_{\rho }\left(\zeta \right),\;\zeta \in M.$$
The structure of the phase space is determined by the symmetry group *G*—a representation that acts irreducibly in the Hilbert space H of the corresponding quantum system [11].

The evolution equation for $W_{\rho }\left(\zeta \right)$ is obtained by mapping the Schrodinger equation into the space of functions on M. To achieve this, a manageable expression for the star-product [4,5,12,13,14,15], e.g., the composition map $\hat{f}\hat{\rho }\to W_{f}\left(\zeta \right)*W_{\rho }\left(\zeta \right)$, is required if $\hat{f}$ is an arbitrary operator acting in H. Unfortunately, the general form for the star-product operation is known only for simplest groups as Heisenberg–Weyl [4,5], SU(2) [16,17,18,19,20] and some generalizations [21]. However, the maps, commonly called the correspondence rules (CR),
(1)$${\hat{c}}_{j}\hat{\rho }\to D_{L}\left({\hat{c}}_{j}\right)W_{\rho }\left(\zeta \right),\;\hat{\rho }{\hat{c}}_{j}\to D_{R}\left({\hat{c}}_{j}\right)W_{\rho }\left(\zeta \right),$$
where ${\hat{c}}_{j}$ are generators of the group *G* and $D^{L,R}\left({\hat{c}}_{j}\right)$ are some differential operators, can be obtained even for more sophisticated groups such as E(2) [22,23] and SU(3) [24]. Explicit expressions for $D_{L,R}\left({\hat{c}}_{j}\right)$ (also known as Boop [25] operators or elements of *D*-algebra [13,14,15,26,27]) are extremely useful as they allow us to obtain the phase-space evolution equations in the case when the dynamics of the system are governed by a Hamiltonian/Lindbladian that is polynomial on the group generators.

The corresponding relations are easily found for the Glauber–Sudarshan *P* and Husimi *Q* functions by using the standard coherent state machinery [26,28,29]. For arbitrary groups, these *P* and *Q* functions can be considered as representatives that are dual to each other of the *s*-parametrized quasidistributions $W_{\rho }^{\left(s\right)}\left(\zeta \right)$ with s=1 and s=−1, respectively. The situation is more involved for the self-dual Wigner function [30,31,32,33,34,35,36], $W_{\rho }^{\left(s=0\right)}\left(\zeta \right)$, which cannot be defined and treated in the same way as $W_{\rho }^{\left(\pm 1\right)}\left(\zeta \right).$ It is precisely the evolution of the Wigner function that represents the main interest due to its sensitivity to the formation of interference patterns and its specific behavior in the semiclassical limit [1,2,3,27,30,31,32,33,34,35,36].

In the present paper, we obtain the correspondence rules for quantum systems possessing SU(1,1) symmetry [37,38,39,40,41,42,43,44,45,46,47] and apply them for the analysis of phase-space dynamics generated by some non-linear (polynomial) Hamiltonians. The classical phase-space in this case is the upper sheet of the two-sheet hyperboloid. Thus, one can distinguish two types of dynamics in such a non-compact manifold: (a) a quasi-periodic evolution, generated by Hamiltonians with a discrete spectrum; and (b) a non-periodic evolution proper to continuous-spectrum systems. We analyze both types of phase-space motion in particular cases of quadratic on the group generators’ Hamiltonians. In addition, we discuss the semiclassical limit of the correspondence rules, focusing on the peculiar dynamical properties of the self-dual Wigner function.

In Section 1, we briefly overview the construction of quasidistribution functions for the SU(1,1) group. In Section 2, the correspondence rules for the Wigner function are obtained. In Section 3, we apply the correspondence rules to deduce the evolution equations for some quadratic on the group generators’ Hamiltonians; we find their exact solutions and analyze the semiclassical limit in Section 4.

## 2. The SU(1,1) Quasidistribution Functions

### 2.1. General Settings

Let us consider a quantum system with the SU(1,1) dynamic symmetry group, living in a Hilbert space H that carries an irrep labelled by the Bargman index $k=\frac{1}{2},1,\frac{3}{2},2,..$, corresponding to the positive discrete series. The group generators form the SU(1,1) algebra satisfying the following commutation relations:(2)$$\left[{\hat{K}}_{1},{\hat{K}}_{2}\right]=-i{\hat{K}}_{0},\;\;\;\;\;\;\;\left[{\hat{K}}_{2},{\hat{K}}_{0}\right]=i{\hat{K}}_{1},\;\;\;\;\;\;\;\left[{\hat{K}}_{0},{\hat{K}}_{1}\right]=i{\hat{K}}_{2}.$$
The Hilbert space H is spanned by the eigenstates of the ${\hat{K}}_{0}$ operator,
(3)$${\hat{K}}_{0}|k,k+m\rangle =\left(k+m\right)|k,k+m\rangle \;,\;m=0,1,\ldots \;,$$
where |k,k〉 is the lowest state of the representation, defined by ${\hat{K}}_{-}|k,k\rangle =0$, ${\hat{K}}_{\pm }={\hat{K}}_{1}\pm i{\hat{K}}_{2}$. The value of the Casimir operator
(4)$$\hat{C}={\hat{K}}_{0}^{2}-{\hat{K}}_{1}^{2}-{\hat{K}}_{2}^{2},$$
is fixed to k(k−1).

Orbits of the state |k,k〉 define a set of coherent states [28]
(5)$$|n\rangle =\cosh^{-2k}\frac{\tau }{2}\sum_{m=0}^{\infty }{\left[\frac{\Gamma (m+2k)}{m!\Gamma (2k)}\right]}^{1/2}e^{-i\phi m}\tanh^{m}\frac{\tau }{2}|k,k+m\rangle \;,\;$$
labeled by the coordinates (τ,ϕ) of hyperbolic Bloch vectors in the upper sheet of the two-sheet hyperboloid
(6)$$n={\left(\cosh\tau ,\sinh\tau \cos\phi ,\sinh\tau \sin\phi \right)}^{⊤}\;,$$
The states (Equation 5) resolve the identity according to
(7)$$\begin{array}{ccc} \hat{I} & = & \frac{2k-1}{\pi }\int d^{2}\;n|n\rangle \langle n|\;, \end{array}$$
(8)$$\begin{array}{ccc} d^{2}n & = & \frac{1}{4}\sinh\tau d\tau d\phi . \end{array}$$
It is convenient to write the overlap of two coherent states in terms of the pseudo-scalar product of the respective Bloch vectors as follows:(9)$$|\langle n|n^{\prime }\rangle {|}^{2}={\left(\frac{1+n\cdot n^{\prime }}{2}\right)}^{-2k},$$
where
(10)$$n\cdot n^{\prime }=\cosh\tau \cosh\tau^{\prime }-\cos\left(\phi -\phi^{\prime }\right)\sinh\tau \sinh\tau^{\prime }.$$
This hyperboloid can be considered as a classical phase space corresponding to our quantum system. Normalized functions f(n)≡f(τ,ϕ) on the hyperboloid can be expanded on the basis of harmonic functions,
(11)$$u_{n}^{\lambda }\left(n\right)={\left(-1\right)}^{n}\frac{\Gamma \left(\frac{1}{2}+i\lambda \right)}{\Gamma \left(\frac{1}{2}+i\lambda +n\right)}P_{-1/2+i\lambda }^{n}\left(\cosh\tau \right)e^{in\phi },$$
as follows:(12)$$\begin{array}{ccc} f(n) & = & \sum_{n=-\infty }^{\infty }\int d\nu \left(\lambda \right)f_{\lambda n}u_{n}^{\lambda }\left(n\right),\;\;\;\;\;f_{\lambda n}=\int d^{2}nf\left(n\right)u_{n}^{*\lambda }\left(n\right), \end{array}$$(13)$$\begin{array}{ccc} d\nu (\lambda ) & = & d\lambda \lambda \tanh(\pi \lambda ) \end{array}$$
The harmonic functions (Equation 11) are eigenfunctions of the Laplace–Beltrami operator $L^{2}$, which is a differential realization of the Casimir operator (Equation 4),
(14)$$L^{2}u_{n}^{\lambda }\left(n\right)=-\left(\lambda^{2}+\frac{1}{4}\right)u_{n}^{\lambda }\left(n\right),$$
where
(15)$$L^{2}={\tilde{k}}_{0}^{2}-{\tilde{k}}_{1}^{2}-{\tilde{k}}_{2}^{2},$$
with
(16)$${\tilde{k}}_{0}=-i\partial_{\phi },\;{\tilde{k}}_{1}=i\sin\phi \partial_{\tau }+i\cos\phi \coth\tau \partial_{\phi },\;{\tilde{k}}_{2}=-i\cos\phi \partial_{\tau }+i\sin\phi \coth\tau \partial_{\phi }$$
being differential realizations of the group generators (Equation 2). The vector field
(17)$$\tilde{k}=\left({\tilde{k}}_{0},{\tilde{k}}_{1},{\tilde{k}}_{2}\right),\;\left[\tilde{k},L^{2}\right]=0,$$
and the Bloch vector n (Equation 6) are orthogonal to each other,
(18)$$n_{0}{\tilde{k}}_{0}+n_{1}{\tilde{k}}_{1}+n_{2}{\tilde{k}}_{2}=0,$$
and satisfy the commutation relations
(19)$$\left[{\tilde{k}}_{j},n_{l}\right]=i\varepsilon_{jlm}n_{m}.$$

### 2.2. s-Parametrized Quasidistribution Functions

The identity resolution (Equation 7) allows us to define $P\left(n\right)=W^{\left(+1\right)}\left(n\right)$ and $Q\left(n\right)=W^{\left(-1\right)}\left(n\right)$ symbols of an operator $\hat{f}$ in the standard form [42,43,44,48,49,50,51,52,53,54,55],
(20)$$Q_{f}\left(n\right)=\langle n|\hat{f}|n\rangle ,$$
(21)$$\hat{f}=\frac{2k-1}{\pi }\int d^{2}nP_{f}\left(n\right)|n\rangle \langle n|,$$
so that
(22)$$\mathrm{Tr}\left(\hat{f}\hat{\varrho }\right)=\frac{2k-1}{\pi }\int d^{2}n\;P_{f}\left(n\right)Q_{\varrho }\left(n\right)\;.$$
It was observed in [45] that all elements of the *s*-parametrized family of quasidistribution functions $W^{\left(s\right)}\left(n\right)$ in the hyperbolic phase space are related to each other through a formal application of a function of the Laplace operator (Equation 15),
$$W_{f}^{\left(s\right)}\left(n\right)={\left[\Phi (L^{2})\right]}^{\frac{s^{\prime }-s}{2}}W_{f}^{\left(s^{\prime }\right)}\left(n\right),$$
where
(23)$$\Phi \left(L^{2}\right)=-\frac{\pi L^{2}}{\cos(\pi \sqrt{1/4+L^{2}})}\prod_{m=1}^{2k-2}\left[1-\frac{L^{2}}{m(m+1)}\right].$$
In particular, the self-dual Wigner symbol is obtained as a “half-way” relation between *Q* and *P* symbols,
(24)$$\begin{array}{ccc} W_{f}\left(n\right) & = & \Phi^{1/2}\left(L^{2}\right)P_{f}\left(n\right)\;=\Phi^{-1/2}\left(L^{2}\right)Q_{f}\left(n\right)\;,\\ Tr(\hat{f}\hat{\varrho }) & = & \frac{2k-1}{\pi }\int d^{2}n\;W_{f}\left(n\right)W_{\varrho }\left(n\right). \end{array}$$
In practice, the application of the $\Phi (L^{2})$ operator is carried out by using the expansions (Equation 12), e.g.,
(25)$$\begin{array}{ccc} W_{\rho }\left(n\right) & = & \frac{2}{\pi }\int d^{2}n^{\prime }\int d\nu \left(\lambda \right)\Phi^{\frac{1}{2}}\left(\lambda \right)\;P_{-\frac{1}{2}+i\lambda }\left(n^{\prime }\cdot n\right)P_{\rho }\left(n^{\prime }\right) \end{array}$$
(26)$$\begin{array}{ccc} & & \frac{2}{\pi }\int d^{2}n^{\prime }\int d\nu \left(\lambda \right)\Phi^{-\frac{1}{2}}\left(\lambda \right)\;P_{-\frac{1}{2}+i\lambda }\left(n^{\prime }\cdot n\right)Q_{\rho }\left(n^{\prime }\right)\;, \end{array}$$
where $P_{-\frac{1}{2}+i\lambda }\left(n^{\prime }\cdot n\right)$ is the conic function [56]; the function Φ(λ) is obtained from the operator (Equation 23) by substituting $L^{2}\to -\left(\lambda^{2}+\frac{1}{4}\right)$ in accordance with (Equation 14) and leading to
(27)$$\Phi \left(\lambda \right)=\frac{{\left(2k-1\right)|\Gamma \left(2k-1/2+i\lambda \right)|}^{2}}{\Gamma^{2}\left(2k\right)},$$
where Γ(z) is the Gamma function.

This also allows us to compute symbols of polynomial functions of the group generators (Equation 2). For instance, taking into account the fact that
(28)$$\begin{array}{ccc} P_{K_{j}}\left(n\right) & = & \left(k-1\right)n_{j}, \end{array}$$
(29)$$\begin{array}{ccc} P_{K_{j}^{2}}\left(n\right) & = & \frac{\left(k-1)(2k-3\right)}{2}n_{j}^{2}\pm \frac{\left(k-1\right)}{2}, \end{array}$$
where the sign “+” is for j=0 and the sign “−” is for j=1,2, one obtains
$$W_{K_{j}}\left(n\right)=\left(k-1\right)\Phi^{1/2}\left(L^{2}\right)n_{j}=\sqrt{k(k-1)}n_{j},$$
and similarly,
$$W_{K_{j}^{2}}\left(n\right)=\frac{\sqrt{k(2k+1)(k-1)(2k-3)}}{3}n_{j}^{2}\pm \frac{k(k-1)}{3}.$$

## 3. Correspondence Rules

### 3.1. Correspondence Rules for Q and P Functions

The correspondence rules (Equation 1) for *P* and *Q* functions are immediately obtained by using the basic properties of the coherent states (Equation 5). In particular, one has the following *D* algebra operators [42,43]: (30)$$\begin{array}{ccc} {\hat{K}}_{j}\hat{\rho } & \to & W_{K_{j}\rho }^{\left(\pm 1\right)}\left(n\right)=D_{L}^{\left(\pm 1\right)}\left({\hat{K}}_{j}\right)W_{\rho }^{\left(\pm 1\right)}\left(n\right), \end{array}$$(31)$$\begin{array}{ccc} \;\hat{\rho }{\hat{K}}_{j} & \to & W_{\rho K_{j}}^{\left(\pm 1\right)}\left(n\right)=D_{R}^{\left(\pm 1\right)}\left({\hat{K}}_{j}\right)W_{\rho }^{\left(\pm 1\right)}\left(n\right),\;j=0,1,2, \end{array}$$
which are convenient to express in vector notation as
(32)$$\begin{array}{c} D_{L,R}^{\left(s\right)}\left({\hat{K}}_{0}\right)=\left(k-\frac{s+1}{2}\right)n_{0}-s\frac{i}{2}{\left(n⋊\tilde{k}\right)}_{0}\pm \frac{1}{2}{\tilde{k}}_{0},\\ D_{L,R}^{\left(s\right)}\left({\hat{K}}_{1,2}\right)=\left(k-\frac{s+1}{2}\right)n_{1,2}-s\frac{i}{2}{\left(n⋊\tilde{k}\right)}_{1,2}\mp \frac{1}{2}{\tilde{k}}_{1,2}\;s=\pm 1, \end{array}$$
where $n_{j}$ and ${\tilde{k}}_{j}$ are the components of the pseudo-Bloch vector (Equation 6) and the vector field (Equation 16), respectively, and the deformed cross-product $n⋊\tilde{k}$ is defined as
(33)$$\begin{array}{ccc} n⋊\tilde{k} & = & \left(n_{1}{\tilde{k}}_{2}-n_{2}{\tilde{k}}_{1},n_{0}{\tilde{k}}_{2}+n_{2}{\tilde{k}}_{0},-n_{0}{\tilde{k}}_{1}-n_{1}{\tilde{k}}_{0}\right), \end{array}$$
(34)$$\begin{array}{ccc} {}[{\tilde{k}}_{j},{\left(n⋊\tilde{k}\right)}_{l}] & = & i\varepsilon_{jlm}{\left(n⋊\tilde{k}\right)}_{m}. \end{array}$$

### 3.2. Correspondence Rules for the Wigner Function

Taking into account the relation (Equation 24), we observe that
$$\begin{array}{ccc} W_{K_{j}\rho }\left(n\right) & = & \Phi^{1/2}\left(L^{2}\right)P_{K_{j}\rho }\left(n\right)=D_{L}^{\left(0\right)}\left({\hat{K}}_{j}\right)W_{\rho }\left(n\right),\\ D_{L}^{\left(0\right)}\left({\hat{K}}_{j}\right) & = & \Phi^{1/2}\left(L^{2}\right)D_{L}^{\left(+1\right)}\left({\hat{K}}_{j}\right)\Phi^{-1/2}\left(L^{2}\right). \end{array}$$
In other words, the elements of the *D* algebra for the Wigner function and *P* functions are related through a similarity transformation generated by the operator (Equation 23). This representation is quite convenient since the vector field (Equation 16) is invariant under the action of the Laplace–Beltrami operator (Equation 15). Transforming the components of the pseudo-Bloch vector (Equation 6) and making use of the orthogonality relation (Equation 18), we arrive at the following form of the CR for the Wigner function (see Appendix A):(35)$$D_{L,R}^{\left(0\right)}\left({\hat{K}}_{j}\right)=\frac{1}{2}\left\{n_{j}A\left(L^{2}\right)-i{\left(n⋊\tilde{k}\right)}_{j}B\left(L^{2}\right)\pm {\tilde{k}}_{j}\right\},$$
where
(36)$$\begin{array}{ccc} A(L^{2}) & = & \frac{1}{2\varepsilon }\Psi \left(L^{2}\right)-\frac{\varepsilon }{2}\Psi^{-1}\left(L^{2}\right),\;B\left(L^{2}\right)=\varepsilon \Psi^{-1}\left(L^{2}\right), \end{array}$$
(37)$$\begin{array}{ccc} \Psi (L^{2}) & = & {\left[2-4\varepsilon^{2}\left(2L^{2}+1\right)+2\sqrt{1-4\varepsilon^{2}\left(2L^{2}+1\right)+16\varepsilon^{4}L^{4}}\right]}^{1/2}, \end{array}$$
and
(38)$$\varepsilon ={\left(2k-1\right)}^{-1}.$$

## 4. Evolution Equations for the Wigner Function

Applying the CR (Equation 35) to linear Hamiltonians, commonly appearing in the description of non-degenerated parametric processes, with a realization in terms of boson operators, ${\hat{K}}_{0}=\left({\hat{a}}^{†}\hat{a}+{\hat{b}}^{†}\hat{b}+1\right)/2$, ${\hat{K}}_{+}={\hat{a}}^{†}{\hat{b}}^{†}$, ${\hat{K}}_{-}=\hat{a}\hat{b}$, [57,58],
(39)$$\hat{H}=\sum_{j=0}^{2}c_{j}{\hat{K}}_{j},$$
we immediately obtain the equation of motion for the Wigner function [37],
(40)$$i\partial_{t}W_{\rho }\left(n\right)=\left(c_{0}{\tilde{k}}_{0}-c_{1}{\tilde{k}}_{1}-c_{2}{\tilde{k}}_{2}\right)W_{\rho }\left(n\right),$$
where the first-order differential operators ${\tilde{k}}_{j}$ are defined in (Equation 16).

In the case of quadratic Hamiltonians,
(41)$$\hat{H}=\chi {\hat{K}}_{j}^{2},$$
the evolution equations take the form
(42)$$i\partial_{t}W_{\rho }\left(n\right)=\pm \chi \left(n_{j}A\left(L^{2}\right)-i{\left(n\times \tilde{k}\right)}_{j}B\left(L^{2}\right)\right){\tilde{k}}_{j}W_{\rho }\left(n\right),$$
where the sign “+” is for j=0 and the sign “−” is for j=1,2.

For instance, the equation of motion for the Hamiltonian describing Kerr-like nonlinearity [59],
(43)$$\hat{H}=\chi {\hat{K}}_{0}^{2}$$
in hyperbolic coordinates (τ,ϕ) is reduced to
(44)$$\partial_{t}W_{\rho }\left(\tau ,\phi \right)=-\chi \left(\cosh\tau A\left(L^{2}\right)+\sinh\tau \partial_{\tau }B\left(L^{2}\right)\right)\partial_{\phi }W_{\rho }\left(\tau ,\phi \right).$$
Equation (Equation 42) admit exact solutions in the following form
(45)$$W_{\rho }\left(n|t\right)=\frac{1}{2\pi }\int d\nu \left(\lambda \right)\int dn^{\prime }\Phi^{-1/2}\left(\lambda \right)P_{-1/2+i\lambda }\left(n\cdot n^{\prime }\right)Q_{\rho }\left(n^{\prime }|t\right),$$
in accordance with relations (Equation 24), where the corresponding $Q_{\rho }\left(n|t\right)\;$ functions in the basis of eigenfunctions of the ${\tilde{k}}_{j}$ operators satisfy some first-order partial differential equations. In Appendix B and Section B.1, we present explicit forms of $Q_{\rho }\left(n|t\right)$ for quadratic Hamiltonians possessing a discrete spectrum (Equation 43) and a continuous spectrum,
(46)$$\hat{H}=\chi {\hat{K}}_{2}^{2},$$
describing effective four-photon processes [60,61]. It is important to stress that Hamiltonians (Equation 43) and (Equation 46) are not unitary equivalent under SU(1,1) transformations and describe qualitatively different evolutions on the hyperboloid.

A comparison of the quantum and semiclassical dynamics is given in the next section.

## 5. Semiclassical Limit

The semiclassical expansion is usually performed over the inverse powers of some physical parameter (the semiclassical parameter), which acquires a large value for a given quantum system prepared in an appropriate initial state. From a mathematical perspective, the semiclassical limit for systems with the SU(1,1) symmetry corresponds to a large Bargman index, as can be observed from Equation (Equation 35). Then, ε defined in Equation (Equation 38) can be considered as a semiclassical expansion parameter whenever ε≪1. In physical realizations, this corresponds to the inverse of the difference of excitations in two-mode interaction Hamiltonians, the inverse coupling constant for the singular oscillator, etc. [28].

It is easy to see that in the semiclassical limit, the operational function (Equation 37) behaves as
(47)$$\Psi \left(L^{2}\right)≃2-\varepsilon^{2}\frac{\left(2L^{2}+1\right)}{2},$$
so that
$$A\left(L^{2}\right)=\varepsilon^{-1}+O\left(\varepsilon \right),\;B\left(L^{2}\right)=O\left(\varepsilon \right).$$
Thus, the zero-order approximation of the CR for the Wigner function (Equation 35) reads as,
(48)$$D_{L,R}^{\left(0\right)}\left({\hat{K}}_{j}\right)=\frac{1}{2}\left(\varepsilon^{-1}n_{j}\pm {\tilde{k}}_{j}\right)+O\left(\varepsilon \right),$$
while for the *Q* and *P* functions, the CRs preserve their original structure (Equation 33).

In particular, the evolution Equation (Equation 42) is reduced to the Liouville form:(49)$$\begin{array}{ccc} \partial_{t}W_{\rho } & = & -\varepsilon^{-1}{\left\{W_{K_{j}^{2}},W_{\rho }\right\}}_{P}+O\left(\varepsilon \right), \end{array}$$(50)$$\begin{array}{ccc} {\left\{f,g\right\}}_{P} & = & \frac{1}{\sinh\tau }\left(\partial_{\phi }f\partial_{\tau }g-\partial_{\tau }f\partial_{\phi }g\right) \end{array}$$
Here, the leading term is a first-order differential operator describing the classical dynamics, and the first-order corrections to the classical motion vanish. According to Equation (Equation 49), every point of the Wigner function evolves along the corresponding classical trajectory $n\left(t\right)=\left(\tau (t),\phi (t)\right)$,
(51)$$W_{\rho }\left(n|t\right)=W_{\rho }\left(n\left(t\right)\right),$$
leading to a deformation of the initial distribution in the course of an anharmonic dynamics. This, so-called Truncated Wigner Approximation [62,63,64,65,66,67,68,69,70,71] has been widely used in quantum systems with different symmetries for the description of short-time dynamic effects.

It is worth observing that the semiclassical parameter is inversely proportional to the representation (Bargman) index, which is consistent with the semiclassical limit of the Berezin–Toeplitz quantization approach [53,54,55]. However, its explicit form is different for every *s*-parametrized quasidistribution $W_{\rho }^{\left(s\right)}\left(n\right)$. For instance, if follows from (Equation 33) that
$$Q_{K_{0}^{2}}*Q_{\rho }={\left(D_{L}^{\left(-1\right)}\left({\hat{K}}_{0}\right)\right)}^{2}Q_{\rho }=Q_{K_{0}^{2}}Q_{\rho }+\frac{{\left(2k+1\right)}^{-1}}{\sinh\tau }\partial_{\tau }Q_{K_{0}^{2}}\partial_{\phi }Q_{\rho }+O\left(k^{-2}\right),$$
which implies that the appropriate semiclassical parameter for the *Q* function is ${\left(2k+1\right)}^{-1}$ instead of ${\left(2k-1\right)}^{-1}$ as for the Wigner function. In particular, the equations of motion for the *Q* and *P* functions expanded in powers of $\varepsilon ={\left(2k-1\right)}^{-1}$ do not acquire the form (Equation 49) in the semiclassical limit, since the first-order corrections to the Poisson brackets would be of order O(1).

In the case of evolution generated by the Hamiltonian (Equation 43), the classical equations of motion,
(52)$$\dot{\tau }=0,\;\dot{\phi }=-2k\chi \cosh\tau ,$$
describe well only the initial deformation (squeezing) of the coherent state (Equation 5) up to times $\sqrt{k}\chi t_{sem}≲1$. The early stage of squeezing of the distribution is followed by the formation of *N*-component Schrodinger cat states at χt=π/N, along with a typical interference pattern, the description of which is beyond the semiclassical approximation. In Figure 1 we plot the semiclassical (Equation 51) and quantum (Equation 45), (Equation 69) evolution of the Wigner function of an initial coherent state (Equation 5) under the action of the Hamiltonian (Equation 43).

The evolution generated by the Hamiltonian (Equation 46) is very different from that induced by (Equation 43). The classical trajectories are obtained from
(53)$$\begin{array}{ccc} \dot{\phi } & = & 2k\chi \sin^{2}\phi \cosh\tau , \end{array}$$
(54)$$\begin{array}{ccc} \dot{\tau } & = & -2k\chi \sinh\tau \sin\phi \cos\phi , \end{array}$$
preserving the integral of motion $E=k^{2}{\left(\sinh\tau \sin\phi \right)}^{2}$. The initial coherent state |τ=0,ϕ=0〉 located at the origin of the hyperboloid suffers a deformation in the vicinity of the minimum of the classical potential (mainly in the valley along the axis $n_{2}$),
(55)$$\langle n|{\hat{K}}_{2}^{2}|n\rangle \approx k^{2}\sinh^{2}\tau \sin^{2}\phi ,$$
according to Equations (Equation 53) and (Equation 54) for $\chi t_{sem}≲1$ at long time scales. In other words, the quantum evolution of the initial distribution corresponding to the coherent state located at the minimum of the potential (Equation 55) is well simulated by semiclassical dynamics. In Figure 2, we plot the semiclassical (Equation 51) and quantum (Equation 45), (Equation 84) evolution of the Wigner function of an initial coherent state (Equation 5) located at τ=0 under the action of the Hamiltonian (Equation 46).The main difference between the semiclassical and the quantum evolutions of the Wigner function is the appearance of small amplitude ripplings and a slight bending toward the axis $n_{1}$ in the latter. Observe that in this case, there is no emergence of the Schrodinger cat states. It is worth noting that the long-time quantum evolution of distributions that are not located initially at the origin of the hyperboloid may significantly differ from its classical counterpart.

## 6. Conclusions

We have obtained the correspondence rules for the *s*-parametrized distributions in the hyperbolic phase space. The relations (Equation 33) and (Equation 35) allow us to deduce the exact evolution equations for polynomial Hamiltonians on the SU(1,1) algebra generators. Those equations can be solved in a systematic way for diagonal quadratic Hamiltonians (Equation 41).

The semiclassical limit corresponds to the large values of the Bargman index, which labels the discrete irreducible representations of the SU(1,1) group. The leading order term of the semiclassical expansion of the evolution equation for the Wigner function is reduced to the Poisson brackets on the hyperboloid. Surprisingly, the exact long-term non-harmonic evolution of certain states generated by the continuous-spectrum Hamiltonian (Equation 46) is well described in the semiclassical approximation (Equation 49). This contradicts our intuition of a typical behavior of phase-space distributions, the evolution of which is governed by non-linear (on the group generators) Hamiltonians, as occurs in case of the discrete-spectrum Hamiltonian (Equation 43), where the emergence of the Schrodinger cat states cannot be explained from the classical point of view.

## Figures and Tables

**Figure 1 entropy-24-01580-f001:**
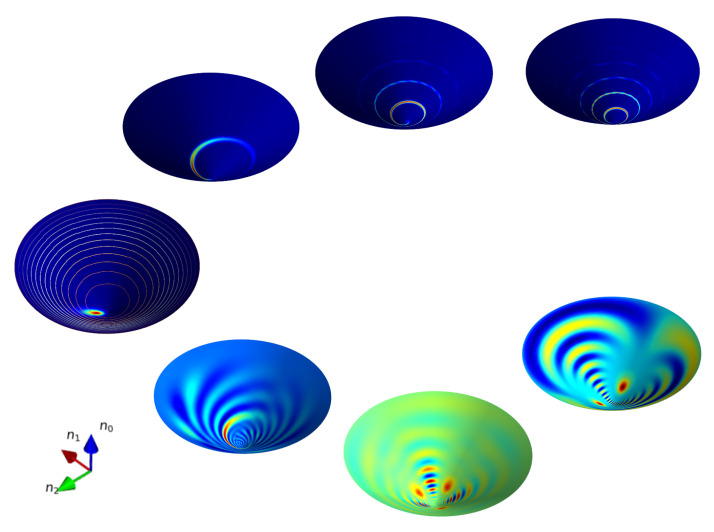
Snapshots of the Wigner function describing the evolution generated by the Hamiltonian $\hat{H}={\hat{K}}_{0}^{2}$ at times t=0,0.2,π/3, π/2 for the initial state |τ=1.5,ϕ=0〉. The upper panel and lower panels describe the semiclassical and quantum dynamics correspondingly.

**Figure 2 entropy-24-01580-f002:**
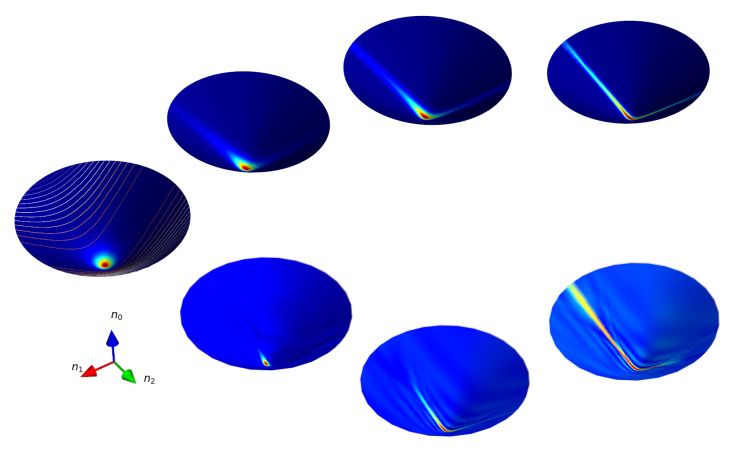
Snapshots of the Wigner function describing the evolution generated by the Hamiltonian $\hat{H}={\hat{K}}_{2}^{2}$ at times t=0,0.2,1,2 for the initial state |τ=0,ϕ=0〉. The upper panel and lower panels describe the semiclassical and quantum dynamics correspondingly.

## Data Availability

Not applicable.

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
