# Peer review of "Correspondence Rules for SU(1,1) Quasidistribution Functions and Quantum Dynamics in the Hyperbolic Phase Space"

_entropy, 2022, doi:10.3390/e24111580_

Round 1

Reviewer 1 Report

The authors systematically derive the explicit differential form for the action of the generators of SU(1, 1) group on the corresponding s-parametrized symbols, and have obtained the correspondence rules for the s-parametrized distributions in the hyperbolic phase-space.

I consider this work a welcome addition to the new results on the SU(1,1) studies and the dynamical properties of the self-dual Wigner function. I recommend publication if the authors consider/answer the following hints/questions in a revised version:

1. Provide the proposals for the physical realization of the Hamiltonians in Eq. (41), (43), (45), and (48);

2. Similar work has introduced su(1,1) Wigner function by using s-parameterized symbols [Klimov et al., aXiv:2012.02993v1, SU(1,1) covariant s-parametrized maps], and used the composition rule of the su(1,1) operators to get the Wigner function [Akhtar et al., Phys. Rev. A 106, 043704 (2022), Sub-Planck phase-space structure and sensitivity for SU(1,1) compass states];

3. Correct the typos, such as, Lindblandian, systematically, … .

Reviewer 2 Report

This paper studies the Wigner function and the semiclassical limit of systems with SU(1,1) symmetry. There is a lot of work in this area, and sections 1-3 and the appendices A,B contain review material which is needed later (but it could be presented more concisely). The material in sections 4,5 and the figures are a novel and interesting contribution to the subject. In section 5 on the semiclassical limit, I would have added something about the Berezin formalism (in the SU(1,1) context).

I recommend that the paper should be published in the special issue of entropy, after the authors consider the optional revisions: reduction of review material and add something about the Berezin formalism in section 5.
